# REVIEW ARTICLE

# Formalising recall by genotype as an efficient approach to detailed phenotyping and causal inference

Laura J. Corbin et al.[#]

Detailed phenotyping is required to deepen our understanding of the biological mechanisms behind genetic associations. In addition, the impact of potentially modifiable risk factors on disease requires analytical frameworks that allow causal inference. Here, we discuss the characteristics of Recall-by-Genotype (RbG) as a study design aimed at addressing both these needs. We describe two broad scenarios for the application of RbG: studies using single variants and those using multiple variants. We consider the efficacy and practicality of the RbG approach, provide a catalogue of UK-based resources for such studies and present an online RbG study planner.

Genome-wide association studies (GWAS) have identified thousands of common genetic variants related to complex traits and diseases[1]. To deepen the understanding of the biological mechanisms underlying specific genetic association results or the impact of potentially modifiable risk factors, new research ideally requires detailed phenotyping and analytical frameworks allowing causal inference. Exhaustive phenotyping in the same discovery collections can be impractical or prohibitively expensive[2] and leads to situations in which measurement precision and quality or proximity to underlying biology is compromised by the use of cheaper pragmatic approaches. There has been substantial growth in the availability of bioinformatic resources able to help break down association results, but less often seen is the explicit use of genetic data to design new studies that could contribute to the understanding of specific association signals, or the impact of potentially modifiable risk factors. Recall-by-Genotype (RbG) studies recall participants, patients or their samples for extensive investigation based on informative genetic variation. These are not standard human genetic association studies, but rather studies that explicitly use existing genetic data as a basis for the design of efficient investigations of mechanism and causality.

In this article, we describe the motivation for and characteristics of RbG studies and why they can be useful for both the examination of specific association results and the efficient extension of applied genetic epidemiology. We discuss the practicalities of incorporating genotypic data into population-based study designs and provide a catalogue of UK-based study resources and an online tool to aid the design of new RbG experiments. Overall, we conclude that RbG studies can help dissect existing genetic associations and make efficient use of the genetic prediction of risk factor exposure through the execution of novel and genotype-informed studies. However, the efficacy of the RbG design depends on a number of study-specific factors and therefore careful consideration should be given as to whether RbG is the optimal design for any given

research question. There is further work to be done by the community in developing protocols and procedures to support RbG studies, in particular to address the potential ethical challenges associated with recruitment by genotype.

**Rationale for genotype-based sampling strategies**. By sampling in an informed manner, targeted studies can be undertaken that allow the examination of dense phenotypic information in sample sizes that are both financially and practically feasible and have the potential to optimise analytical power. Studies that recruit subgroups of participants from the extremes of phenotype distributions (such as lean and obese individuals) have been used in epidemiological investigations for many years; however, these studies suffer well-known limitations of observational epidemiology[3]. In contrast to these, RbG studies use naturally occurring genetic variants robustly associated with specific traits and diseases to stratify individuals into groups for comparison and are novel and beneficial for two reasons. Firstly, by exploiting the key properties of genetic variants that arise from the random allocation of alleles at conception (Mendelian randomization (MR))[3–5], RbG studies enhance the ability to draw causal inferences in population-based studies and minimise problems faced by observational studies (Fig. 1)[6]. Indeed, the often-used comparison of MR to randomised controlled trials (RCTs) is structurally closer for RbG than more conventional applications of this analytical approach[7–9]. Secondly, focusing phenotypic assessments on carefully selected population subgroups can improve insight into mechanism and the aetiology of health outcomes in a cost-efficient manner through targeted deployment of more precise and informative phenotyping across already known biological gradients.

These key features ensure results from RbG studies can be useful in a variety of settings, including in the realm of drug development. For example, data from both GlaxoSmithKline[10]

and AstraZeneca[11] show that genetic target linkage to disease increases the rate at which drugs are approved. Currently, one of the main sources of genetic support are results from GWAS (for example, those in GWASdb[12]) and these seem to be particularly useful in earlier stages of the drug development process[10]. However, the influence of genetic support appears to be less strong in progression from Phase III trials to approval[10], suggesting that there is still progress to be made in refining molecular targets. Furthermore, RbG studies may be able to realise the concept of dose–response curves derived from 'experiments of nature' described by Plenge et al. [13], where naturally occurring mutations can be utilised to estimate the efficacy and toxicity of a drug.

**Exemplars of RbG design in population health studies**. Forms of RbG have appeared in designs looking to optimise RCT and investigate pharmacogenetic relationships[14–18], but have not been fully described for population-based resources. RbG study designs are likely to develop further, but here we present design considerations for RbG in simple form. We split the RbG approach into two categories for the purpose of description; RbG using a single variant (RbG$^{sv}$) and RbG using multiple variants (RbG$^{mv}$). The former can be viewed as a focus on the use of specific (potentially rare and large-effect) loci to understand biological pathways of interest; in contrast, the latter uses polygenic contributions to exposures of interest in study designs more efficiently than conventional MR analyses. These approaches have the same inferential properties based on the properties of genetic data; however, they describe differing analytical scenarios and illustrate the potential variety in this application of human genetics.

RbG$^{sv}$ studies are the most intuitive type of RbG, where strata defined by a single genetic variant are used as the basis for the recall of samples or participants for further phenotypic examination. This type of RbG study may focus on functional variants known to induce a direct biological change; however, genetic variants may also be chosen if they have uncharacterised or predicted effects (i.e., loss-of-function variants, *cis*-regulatory variants or intronic variants that alter DNA-protein binding at potential drug targets)[19]. These variants provide natural experiments able to yield information about the specific role of biological pathways as well as gradients within them and potentially inform on both the safety and the efficacy of medicines. For RbG$^{sv}$ studies, participants or patients or their samples are recruited and phenotypes measured based on genotypic groups in a manner not dissimilar to the arms of a clinical trial. Recall in this way yields groups in which detailed phenotyping can be undertaken to assess the specific impact of a genetic change or the aetiology of an outcome. An early example of this approach was an investigation of the effects of the peroxisome proliferator-activated receptor-γ Pro12Ala polymorphism on adipose tissue non-essential fatty acid metabolism[20]. Further examples of RbG$^{sv}$ are included in Box 1. Additional studies currently underway have had protocols reported in advance of their completion[21, 22].

A form of RbG$^{sv}$, which has received attention in the literature recently, relates to the concept of 'human genetic knockouts', that is, individuals carrying rare homozygous predicted loss-of-function (pLoF) mutations. These are useful in supporting understanding of biological pathways because they come close to simulating the ablation of protein function[23]. By sequencing relatively large numbers of individuals from populations in which homozygous genotypes might be enriched (e.g., founder populations and those with high consanguinity rates), researchers have successfully identified hundreds of pLoF mutations[23]. In their

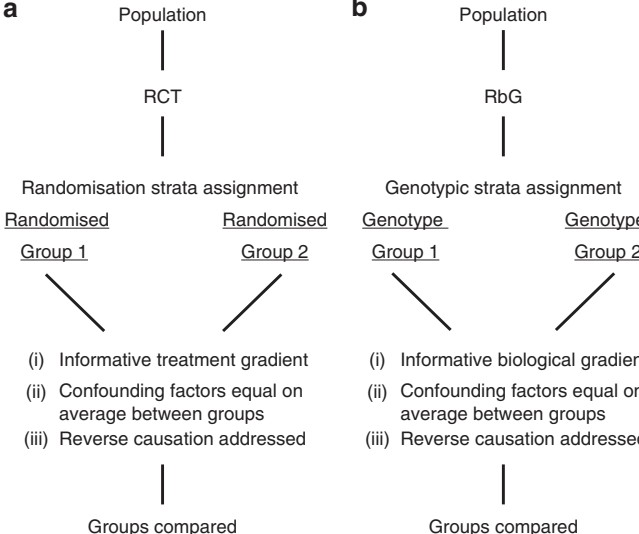

**Fig. 1** Properties of RbG strata compared to randomised control trials. **a** For randomised controlled trials (RCTs), participants are randomly allocated to intervention or control groups. Randomisation should equally distribute any confounding variables between the two groups. **b** For Recall-by-Genotype (RbG) studies, strata are defined by genotype and, analogous to RCTs, potential confounding factors are equally distributed between groups. Hence, RbG studies are not subject to reverse causality or confounding factors with respect to the phenotype under study

## BOX 1: | Examples of RbG studies

**Melatonin signalling and type 2 diabetes**

Several GWASs have identified >100 genetic variants associated with type 2 diabetes (T2D), including a common variant in the melatonin receptor 1b gene (*MTNR1B*). However, the mechanism of how glucose metabolism and development of T2D are affected by melatonin remains elusive. Tuomi et al.[52] demonstrated that rs10830963, an eQTL for *MTNR1B* in human islets, affects insulin release. To test the hypothesis that activation of *MTNR1B* would result in a reduction of glucose-stimulated insulin secretion, Tuomi et al. employed an RbG[sv] study design. Twenty-three non-diabetic individuals with two copies of the risk allele (GG) and 22 individuals with two copies of the non-risk allele (CC) were recruited for the study during which they received 4 mg of melatonin for 3 months. The participants underwent an oral glucose tolerance test before and after 3 months of melatonin treatment and levels of plasma glucose, insulin, glucagon and melatonin were measured. The study found that insulin secretion was inhibited by melatonin treatment, with higher glucose levels in risk allele carriers. Results from this RbG[sv] study suggest that melatonin might be protective against nocturnal hypoglycemia.

**IL2RA polymorphisms and T-cell function**

In type 1 diabetes (T1D), the malfunction of CD4[+] regulatory T cells (Tregs) results in T-cell-mediated autoimmune destruction of pancreatic beta cells. The function of Tregs may be influenced by gene polymorphisms in the IL-2/IL-2 receptor alpha (IL2RA) pathway. Several interleukin-2 (IL-2) receptor alpha-chain (*IL-2RA*) gene haplotypes (rs12722495, rs11594656 and rs2104286) have been shown to be associated with T1D[53, 54]. To investigate whether the *IL-2RA* haplotypes are associated with different expression of IL2RA on the surface of peripheral blood T cells, Dendrou et al.[55] employed an RbG[sv] design, recruiting 50 homozygous or heterozygous individuals for each of the 3 protective haplotypes and 50 homozygous individuals for the susceptible haplotype. Blood samples were collected and the surface expression of IL2RA on peripheral blood T cells measured. Individuals with the protective rs12722495 haplotype in *IL-2RA* had increased expression of IL2RA on the surface of memory CD4[+] T cells and increased IL-2 secretion compared to individuals with the susceptible haplotypes or those with the protective rs11594656 or rs2104286 haplotype. In a second study, Garg et al.[56] employed an RbG[sv] design recruiting healthy individuals according to their genotype at *IL2RA*-rs12722495 to investigate how polymorphisms in *IL2RA* alter Treg function. Blood samples were taken from 34 healthy individuals and T-cell function tested. The study found that the T1D-susceptibility *IL2RA* haplotype correlated with diminished Treg function via reduced IL-2 signalling. Findings from the RbG[sv] studies by Dendrou et al. and Garg et al. informed the design of a successful dose-finding, open label, adaptive clinical trial design of Aldesleukin[57], a recombinant interleukin 2 (IL-2), in participants with T1D to investigate whether Aldesleukin could be potentially used to prevent autoimmune disorders such as T1D by targeting Tregs. The trial found that a single ultra-low dose of Aldeskeukin resulted in early altered trafficking and desensitisation of Tregs, suggesting that Aldeskeukin could be useful to prevent T1D.

study of over 10,000 individuals living in Pakistan, Saleheen et al.[24] identified four participants homozygous for a pLoF variant in the apolipoprotein C3 (*APOC3*) gene, associated with lipid metabolism. By re-contacting one homozygous proband, researchers were able to identify and recruit six pLoF carriers and seven non-carriers from the same family for detailed physiologic examination. Participants underwent an oral fat load followed by serial blood testing for 6 h, which showed pLoF homozygotes had lower post-prandial triglyceride excursions. Features from this work that are more broadly applicable within the RbG[sv] framework include the exploitation of founder populations due to the potential enrichment for highly penetrant large effect variants and the potential to expand recruitment to family members of those identified for recall[25–28].

RbG[mv] designs differ in the formation of their strata. Rather than employing specific loci of known or hypothetical effect, RbG[mv] uses multiple genetic variants to design studies focused on the impact of an exposure of interest (e.g., variable body composition, glycaemic profile or complex disease predisposition). The gains afforded in this type of RbG are not through the balanced recruitment of rare mutations of large effect, but the generation of comparison groups small enough for extremely detailed investigation, but where the risk factor exposure gradient is as marked and powerful as in the analyses of the entire population sample.

Consistent with conventional MR analyses, the choice of genetic variants for RbG[mv] studies relies on the ability of genotypic variation to act as a reliable proxy measure for the exposure of interest. Distinct from genetic prediction, this use of multiple genetic variants as markers for modifiable risk (as in more conventional MR designs) requires strong evidence of reliable association. Single genetic variants associated with complex traits or modifiable risk factors often explain only a small proportion of variance in that trait and a strategy employed

to try and recover some of the consequent lack of power of single variant analyses is to generate aggregate genetic risk scores (GRSs)[29–31]. The use of multiple genetic variants in this way can increase the precision of the causal estimate compared with those derived using separate genetic variants[32]. In contrast to conventional MR, once a GRS is constructed within the study sample targeted for RbG (usually as the sum of allele dosages at risk variants weighted by their beta coefficients obtained from an independent GWAS for the exposure of interest), individuals are ranked based on this score, which is then used to stratify participants for recall (Fig. 2). Actual selection of individuals from extremes of the GRS will be dependent on the number and frequency of the variants forming the score, their effect and the number of participants (or samples) available. In addition, it should be considered that while the average genetic composition of a GRS used to recruit participants will be the same, unlike RbG[sv], the precise allocation of genotype will vary from participant to participant. Despite this, the differences across the genetic stratum will carry the same inferential properties as RbG[sv] and allow for causal inference concerning the risk factor being instrumented[6]. An example of an RbG[mv] study designed to investigate the causal relationship between body mass index (BMI) and cardiovascular health in young adults can be found in preprint form[33] (please note this article has not yet been subject to peer-review). In this study, magnetic resonance imaging-derived measures of cardiovascular health were collected on 418 young adults recruited based on a GRS predicting variation in BMI. Both MR and RbG[mv] analyses indicated a causal role of increased BMI on higher blood pressure and left ventricular mass indexed to height[2,7].

**Statistical power and efficiency in RbG.** Undertaken correctly, power calculations illustrate the conditions in which one would consider using an RbG experiment as an approach as opposed to

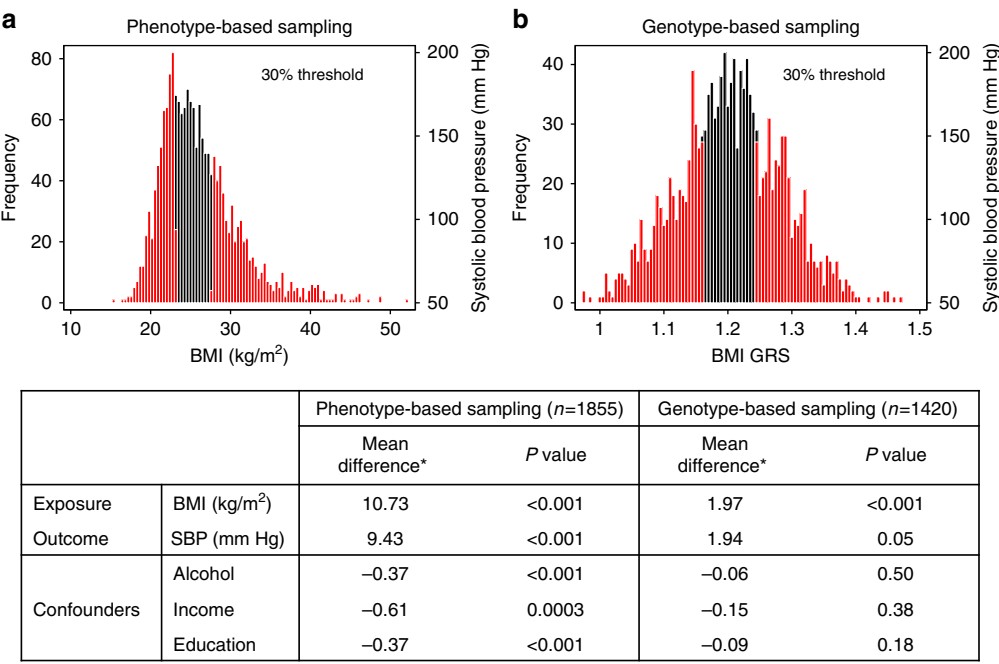

| | | Phenotype-based sampling (*n*=1855) | | Genotype-based sampling (*n*=1420) | |
|---|---|---|---|---|---|
| | | Mean difference* | *P* value | Mean difference* | *P* value |
| Exposure | BMI (kg/m²) | 10.73 | <0.001 | 1.97 | <0.001 |
| Outcome | SBP (mm Hg) | 9.43 | <0.001 | 1.94 | 0.05 |
| Confounders | Alcohol | −0.37 | <0.001 | −0.06 | 0.50 |
| | Income | −0.61 | 0.0003 | −0.15 | 0.38 |
| | Education | −0.37 | <0.001 | −0.09 | 0.18 |

BMI, body mass index; SBP, systolic blood pressure; * difference in means between upper and lower groups

**Fig. 2** Contrast between phenotype and genotype-based sampling strategies. Histograms show the distributions of **a** body mass index (BMI) and **b** the BMI genetic risk score (GRS) in the Avon Longitudinal Study of Parents and Children (ALSPAC). For a description of the ALSPAC data, please see Supplementary Note 2. Red bars represent the top and bottom 30% of these distributions. Mean differences in BMI, systolic blood pressure (SBP) and confounding factors (alcohol, income and education) were compared between the top and bottom 30% of the **a** BMI and **b** BMI GRS distribution. **a** For extreme-phenotype recall studies, participants at the extreme ends of the phenotypic distribution are invited to participate in the study. As an exemplar of this, phenotype data from 1855 individuals in ALSPAC was used. While differences in BMI and SBP are observed between the top and bottom 30% of the BMI distribution, extreme-phenotype sampling strategies are often prone to confounding and potential reverse causality (as shown by the association of the recalled strata with confounding factors). **b** In contrast, RbG studies have the ability to generate reliable gradients of biological difference in combination with essentially randomised groups. As an exemplar of this, genetic data from 1420 individuals in ALSPAC was used to generate a BMI GRS. Differences in BMI and SBP are observed between the top and bottom 30% of the BMI GRS distribution that are not prone to confounding and reverse causality (as shown by the lack of association of the recalled strata with confounding factors)

more conventional sampling methods. Again, it is useful to consider RbG in the RbG$^{sv}$ and RbG$^{mv}$ forms. Power for RbG$^{sv}$ studies can be calculated based on the proposed sample size and the balance of major homozygotes to minor homozygotes/heterozygotes therein (the actual sampling ratio can be adjusted to optimise power as in a conventional case–control design), the phenotypic properties of the outcome measure(s) of interest and the anticipated difference in outcome by recall group. The precise sampling strategy for RbG$^{sv}$ will depend on properties of the target variant and predictions about its mode of inheritance. Here, we consider the implications of recruiting an equal number of major and minor homozygotes (or carriers of the minor allele (heterozygotes) if frequency is very low) in an effort to maximise available biological contrast. However, if it is known, consideration of the appropriate genetic model can aid design (particularly where effects are dominant) and an alternative strategy is to recruit equal (or optimal) numbers of all three genotype groups[34].

A key property of RbG$^{sv}$ design is that study power is independent of the minor allele frequency (MAF) of the target variant; therefore, where random recall designs suffer low power at low MAF, RbG$^{sv}$ does not (Fig. 3a). Consequently, there is most power to be gained at the lower end of the MAF range, where random sampling in relative small samples would fail to yield sufficient numbers of rare variant participants. Despite this, appreciable gains can still be made at moderate MAF if sample size is restricted and/or effect sizes are predicted to be moderate; for example, given a standardized per allele effect of 0.3, a MAF of

0.2 and a sample size of 100, the difference in power between random recall and RbG$^{sv}$ can be over 40%.

Importantly, the efficiency of the RbG$^{sv}$ design comes at some cost as recruiting sufficient participants or samples with low or very low frequency genotypes requires much larger bioresources (with genetic information) from which to recruit (Fig. 3b). For instance, in a study recruiting individuals based on a genetic variant with a MAF of 1% and requiring a total sample size of 50 in each group, the genotyped bioresource would need to contain at least 500,000 individuals in order to identify 50 minor allele homozygotes (assuming Hardy–Weinberg equilibrium). Given that not all participants will be eligible or willing to participate in the RbG study, the required bioresource sample size is likely to be even larger.

Power for RbG$^{mv}$ studies can be considered as a two-part process reflecting not only the properties of the outcome measure, but of the exposure gradient being measured in proxy by the GRS in question. This can be modelled using properties of the genetic variants and their aggregate effects to predict (i) the distribution of the GRS, (ii) the number of participants in the tails of the GRS for any given sample size and (iii) the magnitude of the association between set thresholds of the GRS and the exposure of interest. Given a satisfactory exposure gradient for the GRS in question, the second part of the process follows that of RbG$^{sv}$ studies (i.e., the size of a recall sample to detect biologically informative differences in the outcome phenotype). Again, the efficiency of this approach will be governed by the distribution

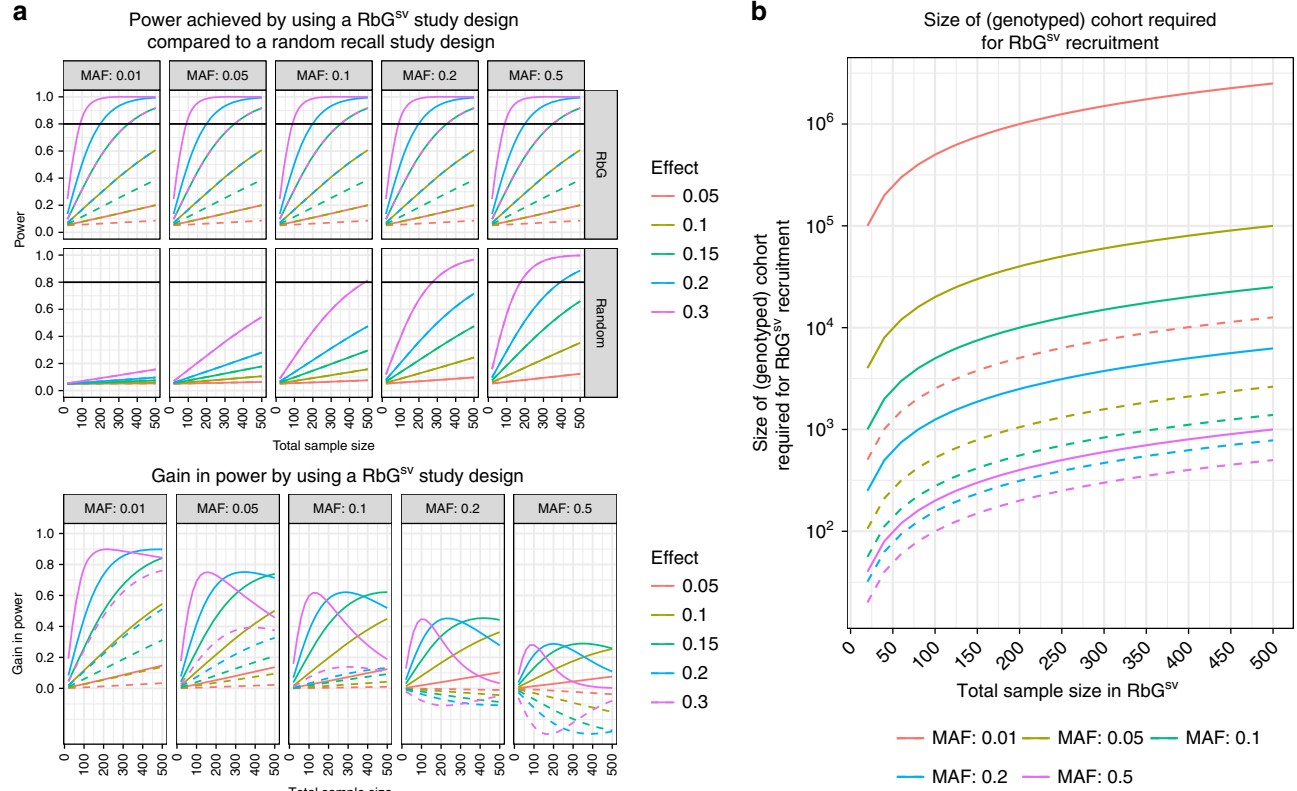

**Fig. 3** Comparative power: RbG^sv versus random recall study design. **a** Top panel: A comparison of power (*y*-axis) achieved by an RbG^sv study design versus a random sample selection design for a given minor allele frequency (MAF) and standardized per-allele effect size. The *x*-axis is the total sample size of the recall experiment. Solid lines represent the situation where an equal number of major and minor homozygotes are recruited. Dashed lines represent the situation where an equal number of major homozygotes and heterozygotes are recruited. Lower panel: A representation of the difference (*y*-axis) between the power within an RbG^sv study design and that from the equivalent random recall experiment. Solid lines represent the situation where an equal number of major and minor homozygotes are recruited. Dashed lines represent the situation where an equal number of major homozygotes and heterozygotes are recruited. **b** An illustration of the expected number of participants with genotypic data (*y*-axis) needed in order to recruit sufficient minor homozygotes or heterozygotes for a given RbG^sv study sample size (*x*-axis) and minor allele frequency (MAF) (assuming HWE and a 100% participation rate). Solid lines represent the situation where an equal number of major and minor homozygotes are recruited. Dashed lines represent the situation where an equal number of major homozygotes and heterozygotes are recruited. For details of how the power calculations were carried out, see Supplementary Note 1. Here we assume a Type I error rate (alpha) of 0.05 and equal-sized genotype groups

and properties of the GRS in question (determining the number of participants at any part of it and the relationship to exposure), but also the practicalities of study-based recruitment (as for RbG^sv).

For RbG^mv studies, power depends on the variance in the exposure explained by the GRS ($R^2_{XG}$), the anticipated relationship between the exposure and the outcome that will be measured ($R^2_{YX}$) (although not likely to be known precisely) and the threshold (percentile) that is used to recruit, as well as sample size (Fig. 4a). The greatest gain in power occurs when the sample groups are recruited from the most extreme part of a GRS distribution, but one must be mindful of the need for large genotyped bioresources from which to recruit in this case (Fig. 4b). As an example, with $R^2_{XG}$ in the range 0.03 (as is currently seen for complex traits such as BMI) and assuming $R^2_{YX} = 0.3$, appreciable power gains (>25%) can be made over random recall using thresholds of between 5 and 20% in samples of 300 or more. Therefore, while the conservative aim for RbG^mv is to achieve equivalent exposure gradient in a smaller sample suitable for extensive investigation, it is evident that for an equivalent outcome power is enhanced (and of course considerable measurement cost savings made).

For both RbG^sv and RbG^mv approaches, there may be situations where power can further be enhanced (and biological effect clarified) when comparing genotype-driven recall groups also group- or pair-matched for characteristics such as age, sex and BMI. Analogous to an RCT, the overall approach in RbG is reliant on the properties of genotype-assigned recall groups, though in certain conditions it may be possible to enhance analyses with appropriate matching strategies. Access to larger sample sizes may reduce the need for matching, but even here matching may be advantageous when there are genotype-driven differences in the potential for ascertainment (e.g., early-onset fatal disease or in selecting non-diabetic individuals for a study of a diabetes risk variant) and this approach has been exercised in existing studies[19, 35]. Other situations that may prompt refinement of the basic RbG design include instances of gene × environment and gene × gene interaction. Though the evidence for consistent examples of these in the literature has been limited to date, in the presence of a gene × environment interaction, for example, the assumption that genotype is orthogonal to all potential confounders may be invalidated due to associations between socioeconomic status and geographic ancestry. Importantly, there remains a danger that efforts to balance or match samples can exacerbate the potential for

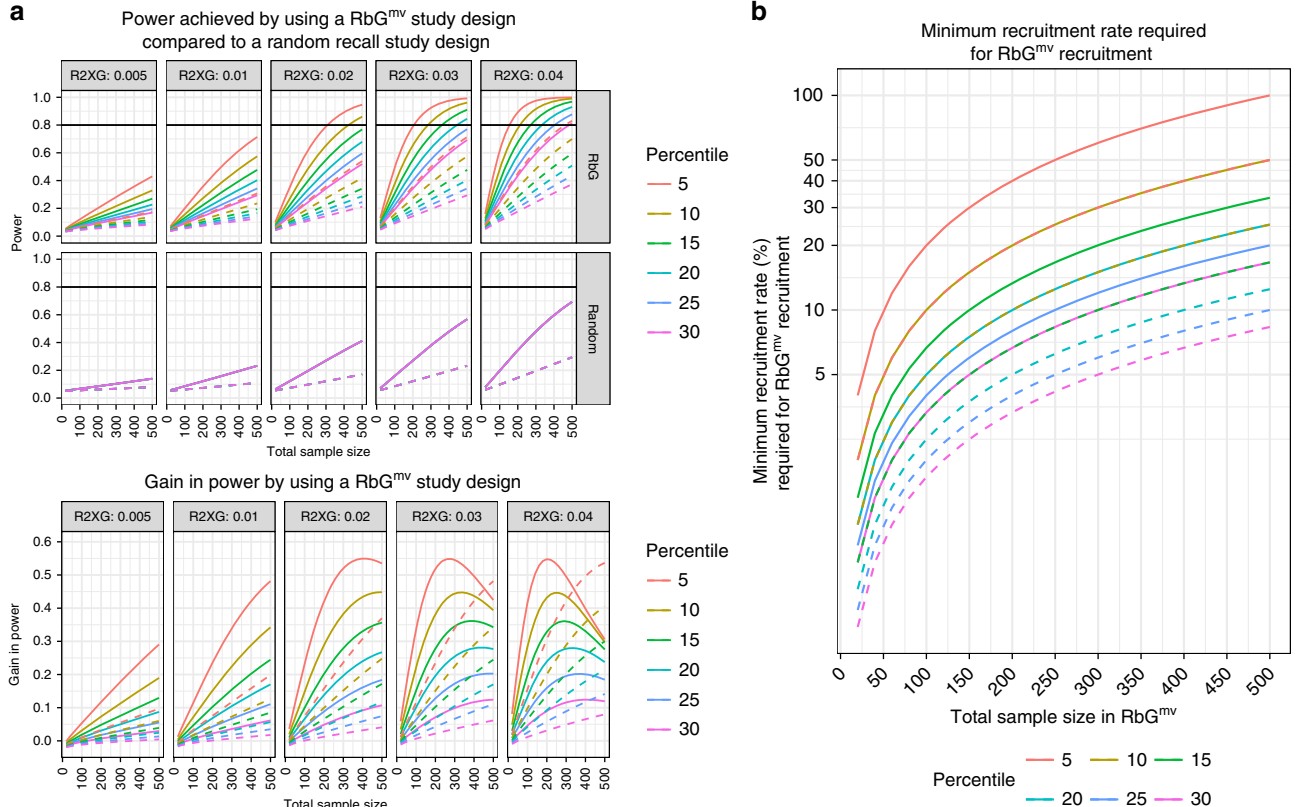

**Fig. 4** Comparative power: RbG$^{mv}$ versus random recall study design. **a** Top panel: A comparison of power (*y*-axis) achieved by an RbG$^{mv}$ study design versus a random sample selection design for a given $R^2_{XG}$ (variance in exposure explained by the genetic risk score (GRS)) and percentile. The *x*-axis is the total sample size. Lower panel: A representation of the difference (*y*-axis) between the power within an RbG$^{mv}$ study design and that from the equivalent random recall experiment. In both the top and bottom panels, solid lines represent the situation where the variance in outcome explained by exposure ($R^2_{YX}$) is equal to 0.3 and dashed lines represent the situation where $R^2_{YX}$ is equal to 0.1. **b** An illustration of the minimum recruitment rate needed in order to recruit sufficient study participants for a given RbG$^{mv}$ study sample size (*x*-axis) and percentile. Solid lines represent the situation where the size of the genotyped cohort (or biobank) is equal to 5000 people and dashed lines represent the situation where the size of the genotyped cohort (or biobank) is equal to 10,000 people. For details of how the power calculations were carried out, see Supplementary Note 1. Here we use the analytical method and assume a Type I error rate (alpha) of 0.05 and equal-sized genotype groups. The 'percentile' is the threshold used to recruit from the GRS distribution in the genotyped cohort (or biobank) in the RbG$^{mv}$ study (e.g., percentile 5 corresponds to recruitment from the top and bottom 5%)

particular types of study bias[36] and the pros and cons of these decisions need to be weighed carefully in study design.

To facilitate the design of RbG experiments based on the scenarios outlined above as 'RbG$^{sv}$' and 'RbG$^{mv}$', we have prepared an online tool for guiding researchers through these steps (see Web resources). The methods used to calculate power for RbG studies are described in more detail in Supplementary Note 1.

**Ethical considerations of RbG.** RbG is a potentially powerful research design, but it creates ethical challenges. The RbG approach is inextricably linked to the issue of disclosing potentially sensitive individual results[37, 38] and places an emphasis on transparency and communication with participants. This of course relates to the nature of both the RbG design and the genetic variation being used to construct the RbG stratum of interest. This is particularly pertinent where potentially penetrant and functional variants are employed in RbG$^{sv}$ designs, but has implications for all forms of RbG. Despite this, there is little published academic work regarding the specific ethical issues in RbG studies.

A small body of literature suggests a need for 'bottom-up research' to be monitored by an independent governance body[39] and that the issues presented with RbG studies are not new but

common to those faced by other approaches, such as the use of medical records[40]. Qualitative data that does exist around this topic compared the experiences of patients (those with the disease of interest) to those of 'healthy volunteers' (recalled from a biobank) following their recruitment on the basis of genotype[41]. This research found that, while patients expressed 'no concerns' about the eligibility criteria, 'healthy volunteers' did not always comprehend the study design or why they had been chosen. This led in some cases to participants assuming a degree of meaningfulness to the genetic data that was unwarranted but, nevertheless, caused them to feel anxious. Seemingly in contrast to this, a qualitative research study in which semi-structured interviews were conducted with 53 young adult participants of the Avon Longitudinal Study of Parents and Children, a cohort of ostensibly 'healthy volunteers' reported that few expressed any immediate concerns about being recruited by genotype[42]. Given that this work has yet to be peer reviewed and is not a systematic analysis (rather excerpts from a small number of interviews), the results of this study must be interpreted with caution. However, the conclusions from this work were that the main reasons for the lack of concern were the trust that participants had developed over their long-term relationship (more than 20 years) with the study, plus a naturally limited knowledge of genetics and modest interest in reported research outcomes. This complements previous research that identified the relationship between

researchers and participants as a factor that may influence how much information is provided, with regular study participants perhaps expecting more under the ethical principles of respect and reciprocity[43]. Although there is clearly scope to expand the body of evidence relating specifically to ethical considerations in RbG, an emerging theme is the responsibility placed onto researchers for the handling of potentially sensitive and disclosive studies.

The very nature of RbG designs highlight a central tension between avoiding the possibility of participant harm through revealing unwanted or misunderstood information and being open and clear when explaining how and why participants are being recruited into studies[37, 38]. In healthy volunteers, it is unlikely that the genetic information used for recruitment to most RbG studies will be either immediately clinically valid or useful, as the precise function of the genetic characteristics will presumably be unknown. However, this does not diminish the need to clearly communicate the study protocol to participants and why they, specifically, have been recruited. To this end, the issue of direct or unwanted indirect disclosure of genotype is of great importance in this type of study. It is of course possible to envisage a situation whereby a threshold of clinical relevance obtained through an RbG study is not reached, but the genetic information could still be of interest to the participant. The employment of sensible mechanisms for assessment of data quality and routes for appropriate feedback (as considered in detail for sequencing studies elsewhere)[44] will clearly be the accepted mode for RbG studies with large effects. However, the issue of addressing a specific genotype-driven effect does serve to illustrate a key advantage of RbG studies over less hypothesis-driven genomic research. It is potentially easier to anticipate the nature of findings for a given recall stratum and therefore the potential relevance of those findings to participants[37, 38].

Related to the nature of the cohort is the extremely important issue of consent and the provision for re-contact of participants within the informed consent process of the original study[41, 45]. While there are a number of 'purpose-built' RbG resources such as The Oxford Biobank, the Exeter 10,000 (EXTEND), the East London Genes & Health (ELGH) and the Extended Cohort for E-health, Environment and DNA (EXCEED) projects whose consent processes deal explicitly with the issue of RbG, in many cases researchers will be looking to recruit from cohort studies established for more general epidemiological research. Therefore, in the event that a network approach to RbG studies is initiated (as described below), careful consideration will need to be given to the extent to which consent and disclosure policies can and should be aligned across studies versus the tailoring of approaches to account for the varied nature of the cohorts involved.

**Resources**. Despite potential advantages of genotype-based sampling strategies, they have so far been underutilised, partly because of limited infrastructure to support them. However, at a time where the potential value of population-based human genetics is being realised in a clinical context[10], recent developments have changed the scientific landscape. A growing number of bioresources have been established or re-purposed to enable RbG studies and are ready for coordinated deployment to maximise RbG designs. In the UK alone, there exists a collection of RbG-ready studies that form a network of genotypic resources and phenotypic expertise suitable for the execution of new studies (see Table 1: UK patient and population-based studies available for RbG studies and the extended version in Supplementary Table 1). A second factor has been the continued fall in genotyping and sequencing costs, which has accelerated discovery and enabled genetic characterisation of large cohorts consented for

RbG studies. Finally, in recent years a number of RbG studies with important findings have been reported that highlight the value of the approach and illustrate key variations on it.

## Future directions, limitations and recommendations

We have presented RbG as a potentially valuable study design in its simplest form within population-based studies. Recall itself is not a novel paradigm to epidemiological studies, where phenotype-driven selection has been a mainstay for the purposes of maximising analytical power. The novelty with RbG comes from the selection process being based on genetic strata, which have the ability to recapitulate biological pathway changes or exposure differences and do so using reliably measured, reproducible and randomly allocated markers. In the correct conditions, this approach has the potential to be both cost-effective and biologically informative.

The ability to measure genetic variation reliably (including that with low MAF) is an important asset to this approach and has been facilitated by both the swathe of GWAS analyses and imputation development that has occurred over the last 5–10 years. However, to take this further, the existence and maturation of effective networks of RbG-ready collections will undoubtedly be required. Not only will these networks allow for the look-up and access of rare variant carriers in reasonable numbers, but local bases of phenotypic expertise will help to develop and exercise the real value of RbG studies in deep phenotyping and enhanced statistical power. For the RbG approach to prove of greatest benefit in the future, this will have to be coupled with large-scale population and patient-based records of genotypic variation data with appropriate consent.

Along with this, there is a series of developments that may enhance the utility of RbG as an approach. Resources are already available that present the possibility of searching the human genome for genetic variants that are particularly suited for use in RbG experiments. Most pertinent to RbG[sv] designs, assessment of variant suitability would likely involve browsing genetic regions of interest for evidence of actual or predicted functional variation using best available data (e.g., the ExAC database[46]) and the marriage of this information to outcome association results and RbG study design parameters. In this way, researchers would be able to conduct a pre-emptive assessment of the likely value and performance of an RbG study. In addition to this, other developments include the formalisation of data-driven recall protocols (where the reduction of extremely complex data for non-hypothesis-driven association signal discovery is followed by deep exploration of results by genotype) and the testing of population-level opt-out strategies (i.e., that avoid disclosure of genotype status—or likely status—with invitation alone) to ensure ethical balance for RbG studies.

There are specific adaptations and potential limitations that are relevant to this approach. Concerning power, current approaches able to assess simplified RbG conditions provide conservative estimates of the performance of RbG studies and need to be developed to further incorporate the application of group and pair-based matching. These techniques are used in RCTs and have the potential to increase statistical efficiency, especially in small sample sizes and where chance or study-specific biases may be present. In addition to refining power calculations and study planning, it is important to consider the potential of employing variants of specific functional effect or sets of genetic variants[47] that act together, interact or are responsible for specific pathway effects. With increasing information about the weight of specific and functional genetic changes and a growing collection of whole genome sequence data available, the opportunity to explore

**Table 1 UK patient and population-based studies available for RbG studies**

| Study | Sample size | Local phenotypic expertise | Patient group/ population sample |
|---|---|---|---|
| The Avon Longitudinal Study of Parents and Children (ALSPAC) | ~9000 (mother child duos) & ~2000 trios. Smaller number of children of index participants (third gen) | Lifecourse epidemiology—birth cohort ('complete' phenotyping) | Population-based cohort |
| East London Genes & Health (ELGH) | 26,476 (at Nov. 2017, actively recruiting, total sample size 100 k) | Human knockouts, primary care e-health records, diabetes and cardiovascular | Population-based cohort (Bangladeshi and Pakistani ethnicity, age > 16) |
| EXtended Cohort for E-health, Environment and DNA (EXCEED) | Over 9300 recruits to date; recruitment planned to continue to 10,000 | Cardiovascular, respiratory, renal, metabolic, infectious disease and cancer | Population-based cohort (aged 30–69) |
| Exeter 10,000 (EXTEND) | 10,000 | Type 2 diabetes, ischaemic heart disease, vascular function and healthy ageing | Population-based sample (based in Exeter; enriched for patients with diabetes; aged > 18) |
| Genetics of Diabetes and Audit Research Tayside Study (GoDARTS) | 9439 cases and 8187 controls | Complete EMR linkage, type 2 diabetes, heart disease, asthma and cancer | Case−control cohort |
| INTERVAL | 50,000 | >6000 molecular phenotypes, including serum NMR metabolomics, plasma MS lipidomics and metabolomics, plasma proteomics, Sysmex FBC, hepcidin and others | Population-based sample of healthy blood donors |
| National Centre for Mental Health | Over 10,000 | Mental health conditions | Population-based cohort (variety of mental health conditions; all ages; primarily Wales-based) |
| The Oxford Biobank | 7900 | Metabolic and anthropometric, obesity | Random, population-based sample of healthy 30–50-year-old men and women (Oxfordshire) |
| Scottish Health Research Register (SHARE) | 50,000 samples obtained. 155,000 consented for spare blood interception | Complete EMR linkage. Type 2 diabetes, heart disease, asthma and cancer. Mobile App Patient Reported Outcomes. | Population-based cohort |
| Generation Scotland: Scottish Family Health Study (GS:SFHS) | 20,032 | Complete EHR linkage, urinary traits and kidney disease, eye phenotypes, family based data analysis | Family-based population cohort |

NMR, nuclear magnetic resonance; MS, mass spectrometry; EHR, electronic health record; EMR, electronic medical records; FBC, full blood count. An expanded version of this table with additional information can be found in Supplementary Table 1

predicted effects in specific clinical scenarios is also increasing[10] and can be extended with RbG studies.

In line with more conventional MR analyses based on non-selected population samples, the quality and nature of the genetic variants used for stratum formation will directly affect the ability to draw inference. Population stratification, genuine (or horizontal) pleiotropy and consequent unanticipated instrument properties have been observed elsewhere[48, 49] and may affect RbG through the invalidation or complication of genetic instruments. Horizontal pleiotropy[3, 50] specifically is a complication that should be viewed in the context of the type of RbG being undertaken however and is an issue that is pertinent to MR more generally[4]. In the case of RbG$^{sv}$, while pleiotropy may complicate the inference drawn from differences between recalled groups, one of the merits of using RbG for single variants will be to explore the potentially diverse and complicating nature of genetic associations validated for health outcomes. For RbG$^{mv}$, the situation is different and while it is theoretically possible for directional (unbalanced) pleiotropy to potentially bias estimates drawn from groups defined by many genetic variants, with increasing numbers of these genetic proxies for complex exposures or risk factors of interest, the likelihood of

this problem decreases[50]. This does not remove pleiotropy as a potential complication (indeed it may be unlikely to ever have a single genetic predictor not involved in complex regulation), but it presents RbG as an approach to explore and account for pleiotropy.

Lastly, unbalanced loss to follow-up by genotype (due to death or behaviour) and the practicalities of study-specific recruitment are potentially limiting factors that need to be considered when undertaking RbG. These limitations can have an impact on the outcomes of this type of design and will benefit from the study of recruitment dynamics in large-scale prospective studies[51]. Overall, as is the case for other forms of MR, these limitations highlight the role of RbG as only part of a required triangulation and replication of evidence when asserting causality or mechanism.

Considering RbG as a vehicle for undertaking detailed and causal dissection of genetic effects and the efficient exploration of potentially causal risk factors, there are recommendations that come from early experiences with studies of this design. These recommendations are presented in Box 2. Overall, RbG study designs have the potential to offer independent and informative biological gradients over which specifically designed studies can interrogate the detailed architecture of confirmed associations. In

---

**Box 2: | Recommendations for RbG implementation**

1. **RbG designs are not appropriate for all studies**. Depending on the nature of the genetic variation in question, the sample type or participant recruitment opportunities and the outcomes of interest, there will be optimal conditions for either RbG$^{sv}$ or RbG$^{mv}$ study designs. These should be carefully considered before undertaking a new RbG investigation. We have described both the types of experiment that may be facilitated by this approach and provide an online application to aid the planning of future work.

2. **Genetic variant(s) should be well characterised**. The genetic variation forming the recall strata are the fundamental building blocks of this study design and the most likely reason that such a study would fail. As for all MR analyses, the limitations to the genetic instrumentation of pathways or risk factors of interest will dictate any inference drawn from analyses undertaken. The integrity of the genetic signal motivating the study (in either RbG$^{sv}$ or RbG$^{mv}$) should be thoroughly assessed.

3. **The full financial and non-financial costs of undertaking an RbG study should be considered**. A deep phenotyping exercise based around an RbG design may yield a definitive single hypothesis answer, but the utility of the sampling frame will be limited by that specific study design. This does not render the by-product resources useless (given the randomised nature of their strata), but this property needs thought.

4. **Transparency, communication (including appropriate disclosure) and thoughtful process in working with participants in RbG studies are paramount**. This is a relatively novel approach for using genetic data and, while the paradigm is simple, the implications are often not. Researchers should consider not just the implications of genotypes employed in this type of study, but also the implications of inviting participants before it is done.

5. **Network RbG studies may be an answer**. The issues of allele frequency, optimising phenotypic expertise, standardizing consent and strategy and reducing the complexity of original study initiation remain and may be best addressed by combining resources. Studies do exist that are suitable for RbG and the federated use of these as a network of RbG resources has the potential to overcome specific RbG limitations.

---

tandem with the driving forces of larger hypothesis-free association studies, the presence of directed follow-up and causal investigation may provide the opportunity to convert some of these outputs into targets for clinical use and future development.

## Web resources

To facilitate the design of RbG experiments based on the scenarios outlined in this paper, we have prepared an online tool for guiding researchers through these steps that is available on the MRC IEU Software Page at: http://www.bristol.ac.uk/integrative-epidemiology/faciliitiesresources/software/ (under 'RbG Study Planner'). The methods used to calculate power for RbG studies are described in more detail in Supplementary Note 1.

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

## Acknowledgements

This work was supported by the Medical Research Council MC_UU_12013/3 (N.J.T., L.J.C., K.H.W., D.A.H.) and MC_UU_12013/1 (G.D.S.). N.J.T. is a Wellcome Trust Investigator (202802/Z/16/Z) and works within the University of Bristol NIHR Biomedical Research Centre (BRC). N.J.T. and V.Y.T. are supported by the CRUK Integrative Cancer Epidemiology Programme (C18281/A19169). The MRC/BHF Cardiovascular Epidemiology Unit is supported by the UK Medical Research Council (MR/L003120/1), British Heart Foundation (RG/13/13/30194) and NIHR Cambridge Biomedical Research Centre. D.S.P. is supported by the BHF Cambridge Centre of Excellence (RE/13/6/30180) and the Wellcome Trust (105602/Z/14/Z). C.M.L. is supported by the Li Ka Shing Foundation and NIHR Oxford Biomedical Research Centre. Work undertaken by P.W.F. related to this manuscript is supported by the European Research Council (ERC-2015-CoG-681742-NASCENT) and the Swedish Research Council (Distinguished Young Researcher Award in Medicine). The EXCEED study at the University of Leicester has been supported by the Medical Research Council (G0902313) and received partial support from NIHR; the views expressed are those of the authors and not necessarily those of the NHS, the NIHR or the Department of Health. The EXCEED study gratefully acknowledges the support of all participants and staff who have contributed to the study. L.V.W. holds a GlaxoSmithKline/British Lung Foundation Chair in Respiratory Research. M.D.T. holds a Wellcome Trust Investigator Award (WT 202849/Z/16/Z). C.J. holds a Medical Research Council Clinical Research Training Fellowship (MR/P00167X/1). M.I.M. is a Wellcome Trust Senior Investigator and an NIHR Senior Investigator. Research support relevant to this manuscript comes from Wellcome Trust (090532, 098381, 106130), Medical Research Council (MR/L020149/1) and NIH (R01DK098032; U01DK105535). The research was supported by the National Institute for Health Research (NIHR) Oxford BRC. The views expressed are those of the author(s) and not necessarily those of the NHS, the NIHR or the Department of Health. Avon Longitudinal Study of Parents and Children (ALSPAC): We are extremely grateful to all the families who took part in this study, the midwives for their help in recruiting them, and the whole ALSPAC team, which includes interviewers, computer and laboratory technicians, clerical workers, research scientists, volunteers, managers, receptionists and nurses. ALSPAC mothers were genotyped using the Illumina human660W-quad array at Centre National de Génotypage (CNG) and genotypes were called with Illumina GenomeStudio. The UK Medical Research Council and the Wellcome Trust (Grant ref.: 102215/2/13/2) and the University of Bristol provided core support for ALSPAC. The authors also acknowledge Professor John Henderson for providing helpful comments on earlier drafts of the manuscript and Matthew Lee for proofreading the final submission.

## Author contributions

N.J.T. conceived this work; L.J.C., V.Y.T., D.A.H., K.H.W. and N.J.T. wrote first drafts and major components of this paper and related applications and material; D.S.P., K.E.T., F.D., T.F., M.D.T., L.V.W., G.D.S., D.M.E., F.K., C.M.L., M.I.M., J.D., M.O'D., M.J.O., S. O'R. and P.W.F. all contributed substantially to the development of this research and the writing of the manuscript; and G.D.S., F.B., J.M.H., M.W.J., C.J., N.K., C.N.A.P., E.R.P., R.A.S., D.A.vH. and J.W. all contributed to the development of this manuscript and to the writing.

## Additional information

**Competing interests:** TF has consulted for Boehringer Ingelheim and Sanofi and received research funding fromGSK. The remaining authors have no conflicts of interest.

Laura J. Corbin [1,2], Vanessa Y. Tan[1,2], David A. Hughes[1,2], Kaitlin H. Wade[1,2], Dirk S. Paul [3,4], Katherine E. Tansey[5], Frances Butcher[6], Frank Dudbridge[7], Joanna M. Howson[3], Momodou W. Jallow[8,9], Catherine John[7], Nathalie Kingston[10], Cecilia M. Lindgren[11,12,13,14], Michael O'Donavan [15], Stephen O'Rahilly [16], Michael J. Owen [15], Colin N.A. Palmer [17], Ewan R. Pearson[17], Robert A. Scott[18], David A. van Heel [19], John Whittaker[8,20], Tim Frayling[21], Martin D. Tobin [7,22], Louise V. Wain [7,22], George Davey Smith[1,2], David M. Evans[1,2,23], Fredrik Karpe[24,25], Mark I. McCarthy [12,24,25], John Danesh[3,4,26,27], Paul W. Franks[24,28,29,30] & Nicholas J. Timpson[1,2]

[1]MRC Integrative Epidemiology Unit at University of Bristol, Bristol, BS8 2BN, UK. [2]Population Health Sciences, Bristol Medical School, University of Bristol, Bristol, BS8 2BN, UK. [3]MRC/BHF Cardiovascular Epidemiology Unit, Department of Public Health and Primary Care, University of Cambridge, Cambridge, CB1 8RN, UK. [4]British Heart Foundation (BHF) Centre of Excellence, Division of Cardiovascular Medicine, Addenbrooke's Hospital, Cambridge, CB2 0QQ, UK. [5]Core Bioinformatics and Statistics Team, College of Biomedical & Life Sciences, Cardiff University, Cardiff, CF10 3XQ, UK. [6]Oxford School of Public Health, University of Oxford, Oxford, OX3 7LF, UK. [7]Department of Health Sciences, University of Leicester, Leicester, LE1 7RH, UK. [8]Department of Epidemiology and Population Health, London School of Hygiene and Tropical Medicine, London, WC1E 7HT, UK. [9]MRC Unit The Gambia (MRCG), Atlantic Boulevard, Fajara, P.O. Box 273, Banjul, The Gambia. [10]National Institute for Health Research (NIHR) BioResource for Translational Research in Common and Rare Diseases & NIHR BioResource Centre Cambridge, University of Cambridge, Cambridge, CB2 0QQ, UK. [11]Big Data Institute at the Li Ka Shing Centre for Health Information and Discovery, University of Oxford, Oxford, OX3 7FZ, UK. [12]Wellcome Trust Centre for Human Genetics, University of Oxford, Oxford, OX3 7BN, UK. [13]Program in Medical and Population Genetics, Broad Institute, Cambridge, MA 02142, USA. [14]NIHR Oxford Biomedical Research Centre, OUH Hospital, Oxford, OX4 2PG, UK. [15]MRC Centre for Neuropsychiatric Genetics and Genomics, Cardiff University, Cardiff, CF24 4HQ, UK. [16]Metabolic Research Laboratories, Institute of Metabolic Science, University of Cambridge, Cambridge, CB2 0QQ, UK. [17]Medical Research Institute, University of Dundee, Ninewells Hospital and Medical School, Dundee, DD1 9SY, UK. [18]Quantitative Sciences, GlaxoSmithKline, Stevenage, SG1 2NY, UK. [19]Blizard Institute, Barts and The London School of Medicine and Dentistry, Queen Mary University of London, London, E1 2AT, UK. [20]Statistical Genetics, Projects, Clinical Platforms, and Sciences (PCPS), GlaxoSmithKline, Research Triangle Park, NC 27709 USA. [21]Genetics of Complex Traits, Institute of Biomedical and Clinical Science, University of Exeter Medical School, Royal Devon and Exeter Hospital, Exeter, EX1 2LU, UK. [22]NIHR Leicester Biomedical Research Centre, Glenfield Hospital, Leicester, LE3 9QP, UK. [23]The University of Queensland Diamantina Institute, The University of Queensland, Translational Research Institute, Brisbane, QLD 4072, Australia. [24]Oxford Centre for Diabetes, Endocrinology and Metabolism, Radcliffe Department of Medicine, University of Oxford, Oxford, OX3 7LE, UK. [25]NIHR Oxford Biomedical Research Centre, Churchill Hospital, Oxford, OX3 7LE, UK. [26]Department of Human Genetics, Wellcome Trust Sanger Institute, Wellcome Trust Genome Campus, Hinxton, CB10 1HH, UK. [27]NIHR Blood and Transplant Research Unit in Donor Health and Genomics, Department of Public Health and Primary Care, University of Cambridge, Cambridge, CB2 0SR, UK. [28]Department of Clinical Sciences, Genetic and Molecular Epidemiology Unit, Clinical Research Centre, Lund University, Skåne University Hospital, Malmö, SE-205 02, Sweden. [29]Department of Public Health and Clinical Medicine, Section for Medicine, Umeå University, Umeå, 907 37, Sweden. [30]Department of Nutrition, Harvard T.H. Chan School of Public Health, Boston, MA 02115, USA

