## [Peer Review File · Nature Communications]

Reviewer #1 (Remarks to the Author):

Corbin and al. examined study design considerations for a novel approach called Recall by Genotype (RbG). They describe two scenarios for RbG: single variant and multiple variants. They consider the efficiency, added value or traditional cohort designs, and practicalities and limitations for incorporating genotypes into such recruitment strategies to build population cohorts.

This is a novel study as it focuses on an exciting area of direction for epidemiology which involves genotype first strategies to recruit individuals. RbG will increasingly be more important in the field of epidemiology / genetic epidemiology / human genetics because of the ability to target certain individuals that have genetic perturbation and then perform follow-up phenotyping, collection of other biological samples such as ipsc, RNA seq, and other OMICS collections of these individuals. Also there has been little published material on study design considerations for RbG so far, and therefore, a discussion of this is welcome.

I have a few thoughts and comments:

1) A major drawback of the RbG approach, though exciting, is that it can be laborious, time-consuming, and expensive because of the need to call back specific individuals. Recall rates can vary widely depending on the nature of deep phenotyping re. associated risks, time commitments to the individual being recalled. It would be helpful if there was a discussion and table of results for previous RbG studies that experienced the rates of recalling individuals (just as in epidemiology studies that discuss participation and drop out rates.

2) There should be a discussion of a recent, high profile paper by Danish Saleheen et al. Nature. 2017 "Human knockouts and phenotypic analysis in a cohort with a high rate of consanguinity" where they recruited APOC3 homozygous KOs from a family and challenged them with oral fat load. This would appear to be a good example of the potential of using RbG to recruit families with an extremely rare founder mutation.

3) I think horizontal pleiotropy will be a much bigger problem than the authors suggest for RbG studies. There is emerging evidence that suggests that pleiotropy is pervasive amongst both GWAS loci and sub-genome-wide significant genetic variants. The authors state that in the paper "horizontal pleiotropy is a complication likely to be more relevant for RbGmv". This might be true if one has multiple instruments for RbGsv and there is a single variant amongst the multiple variants that has little horizontal pleiotropy. However, if pleiotropy is pervasive (as emerging evidence in the medical genetics community suggests), then its less likely to be able to find the rare instance where horizontal pleiotropy does not occur. Also, I think GRS can also be highly pleiotropic depending on which genetic variants are included. I think a more measured discussion of horizontal pleiotropy, and its impact on RbG would be helpful.

4) The RbG approach would appear to benefit from the class of genetic variants that are most damaging re. protein-coding variation particularly LOF carriers, and then deleterious missense variants. For the RbGmv, I think class of genetic variation is less so since it involves multiple genetic variants with small effects collectively. A discussion of how one would prioritize recalling individuals based on their genotypes (do you look for LOF carriers first?) is helpful.

5) Following point 5 above, collection of LOF carriers and deep phenotyping has been discussed in the form of a "human KO project".

(<http://www.nature.com/nrd/journal/v16/n8/full/nrd.2017.139.html>) . Recognition of this specialized form of RbG is warranted.

6) Following point 6 above, if one focuses on LOF recall, then there may only be N=1 individual to recall. In a N=1 study, the LOF carrier genotype is extremely rare but valuable.

7) I like the measured discussion of RbG not being applicable for all genes and that it depends on the hypothesis. One major area of interest is targeting drug discovery genes which I would welcome a discussion.

8) It's unclear if in the RbGmv design that the authors are proposing utilizing all SNVs in the human genome or just genome-wide significant SNVs for a particular trait or locus for the GRS calculation. If it's the former, then this technically is not Mendelian randomization or should be used as causal inference estimation (line 160 - 170) since utilizing all SNVs in the human genome (including ones with tiny effects with P-values ~ 0.05) also includes some SNVs that are definitively associated with the trait of interest (there will be a lot of SNVs with false positives).

Reviewer #2 (Remarks to the Author):

This well-written paper provides a discussion of technical and ethical aspects of study designs involving recall by genotype, for the purpose of detailed phenotypic or causal inference. The paper would benefit from further clarification on two points:

1-Recall process. The authors appropriately note that recalling by genotype raises ethical concerns about participant harm, based on potential misunderstanding of the genotype for which the participant was recruited. This issue, as they note, is particularly delicate if the study is intended to address substantial uncertainties about the meaning of the genotype. Potential participants need transparent information about study procedures and goals but also communication that minimizes the risk that they will misunderstand and potentially over-estimate the implications of their genotype. They might wish to comment on ways in which empiric studies of communication might help to clarify the most appropriate approaches to address this challenge. As a related issue, and potentially more concerning for some participants, they might comment on the ethical concerns related to obtaining the genotypic information that identifies eligible participants. They note that "large genotyped biosources" are needed to enable the recall by genotype studies, and note also that increasing use of genomic assessment in clinical studies and increased use of bio repositories will facilitate this approach. They do not discuss the fact that participants may not always be aware of the scope of genomic analysis available from their samples, particularly if genotyping has been done on stored samples or sample obtained as "clinical waste." Ethical issues that call for more attention include the nature of the initial consent process - e.g., how fully have potential participants been informed about the likelihood that they will be approached for studies of this kind? - and the need for researchers to establish what prior information has been provided to participants about the nature of research with their samples, prior to contact. In some instances these issues will have been addressed proactively, but how should researchers address them if they have not?

2-Appropriateness of controls/biased sampling- The authors discuss selection of comparison groups by genotype, a methodological issue which is fundamental for this kind of research. They also comment in passing about pari-matching for age, sex and BMI. However, given the many indications we have of gene-environment interactions and of the impact of social environments on health outcomes that are also influenced by genotype, it would seem that researchers should also consider other variables relevant to their study question. This methodological point does not seem to have been considered. It may be particularly important given a likely socioeconomic bias in most large genotyped resources.

Reviewer #3 (Remarks to the Author):

In this paper, Corbin and colleagues discuss the motivation for and characteristics of a Recall-by-Genotype (RbG) study design for following up genetic association results. They describe two settings for employing a RbG - one to follow-up single variant associations and a second that

utilizes multiple variants in a risk score type setting. As acknowledged by the authors, the underlying concept behind the RbG design is not new, having been used in epidemiology for many years – that is, recruitment by exposure status. The concept is not particular new for genetic studies either. The authors' intent in this paper is to provide a conceptual framework to this approach.

Overall, I think the motivations for the RbG studies are relatively straightforward and are appropriately spelled out in this article. There are a few areas where I think the authors can elaborate:

1. Founder populations constitute a particularly strong niche for RbG studies because of the potential enrichment for highly penetrant, large effect variants that can provide insights into biology. For example, in the Amish population the RbG design has been used for some time to gain biological insights by calling back selected subjects for more detailed phenotyping (e.g., APOC3: Pollin et al., *Science*, 2008; ABCG8: Horenstein et al; *Arteriosler Thromb Vasc Biol*, 2013; LIPE: Albert et al., *New Engl J Med*, 2014; COL1A2: Daley et al., *J Bone Miner Res*, 2009; SLC30A8: Maruthur et al., *Diabetologia*, 2015) An added feature of the RbG design that the authors might note is the potential not just to call back prior study participants with genotypes of interest, but also to expand recruitment around family members of these individuals.

2. Consent issues and ethical issues: One point that is not explicitly made is that there may be IRB barriers for some studies from recalling individuals based on genotype since this requires disclosure of genotype information, and study subjects may not have been

Many thanks to all the reviewers for their thoughtful comments. Please find below our responses in bold type and excerpts from the paper where necessary, following each of the reviewer points. We can also provide a word document with tracked changes on request (when I tried to upload it as part of the submission it got converted to a pdf and the tracking information was lost).

Reviewers' comments:

Reviewer #1 (Remarks to the Author):

Corbin and al. examined study design considerations for a novel approach called Recall by Genotype (RbG). They describe two scenarios for RbG: single variant and multiple variants. They consider the efficiency, added value or traditional cohort designs, and practicalities and limitations for incorporating genotypes into such recruitment strategies to build population cohorts.

This is a novel study as it focuses on an exciting area of direction for epidemiology which involves genotype first strategies to recruit individuals. RbG will increasingly be more important in the field of epidemiology / genetic epidemiology / human genetics because of the ability to target certain individuals that have genetic perturbation and then perform follow-up phenotyping, collection of other biological samples such as ipsc, RNA seq, and other OMICS collections of these individuals. Also there has been little published material on study design considerations for RbG so far, and therefore, a discussion of this is welcome.

Thank you for your kind appraisal of our manuscript.

I have a few thoughts and comments:

1) A major drawback of the RbG approach, though exciting, is that it can be laborious, time-consuming, and expensive because of the need to call back specific individuals. Recall rates can vary widely depending on the nature of deep phenotyping re. associated risks, time commitments to the individual being recalled. It would be helpful if there was a discussion and table of results for previous RbG studies that experienced the rates of recalling individuals (just as in epidemiology studies that discuss participation and drop out rates.

This is very good point, though it should be held in mind (as outlined in the current draft) that actually recalling participants is only one mode of RbG. The directed recruitment of samples or even data are designs where the laborious nature of participant recruitment is avoided. The former of these has obvious benefits in the opportunity to undertake extremely detailed phenotypic analysis on existing samples in a manner educated by genotype. In the latter case, this may seem circular, however there is an increasingly valuable position in revisiting data according to genotype groups where extremely data heavy and intensive record sets (e.g. brain scanning data such as that in UKBiobank) is collapsed for primary analyses and the purpose of signal discovery, but requires specialist dissection and explosion for follow-up analyses.

Out of these situations, we completely agree with the reviewer, recalling participants is sometimes (and in a study dependent manner) tricky and the basic challenge can be heightened when participants are faced with genetics data. We have tried to comment on this through our discussion of ‘Statistical power and efficiency in RbG’ (i.e. accounting for recruitment) (L238-245) and also touch on the issue in the ‘ethics’ and ‘futures’ (L425-427) sections of the manuscript. Furthermore, we have conducted qualitative work outlining the participant perception of these studies (currently in bioRxiv - <https://www.biorxiv.org/content/biorxiv/early/2017/04/05/124636.full.pdf> and submitted). In most instances (unfortunately), recruitment rates for RbG studies are not generally published and we do not have enough data for a table. Indeed, in our own experience we have seen recruitment rates vary considerably depending on the nature of the experiment and the participants involved. We have attempted to illustrate this in the paper but don’t believe we can provide reliable evidence on a generic RbG recruitment rate at present.

2) There should be a discussion of a recent, high profile paper by Danish Saleheen et al. Nature. 2017 “Human knockouts and phenotypic analysis in a cohort with a high rate of consanguinity” where they recruited APOC3 homozygous KOs from a family and challenged them with oral fat load. This would appear to be a good example of the potential of using RbG to recruit families with an extremely rare founder mutation.

We thank the reviewer for this point and apologies for the omission which was a function of timing. We completely agree and as a result have added a paragraph into the “Exemplars ...” section (L163-179).

“A form of RbGsv, which has received attention in the literature recently, relates to the concept of “human genetic knockouts”, that is, individuals carrying rare homozygous predicted loss-of-function (pLoF) mutations. These are useful in generating understanding of biological pathways because they come close to simulating the ablation of protein function²³. By sequencing relatively large numbers of individuals from populations in which homozygous genotypes might be enriched (for example, founder populations and those with high consanguinity rates) researchers have successfully identified hundreds of pLoF mutations²³. In their study of over 10,000 individuals living in Pakistan, Saleheen et al. (2017)²⁴ identified four participants homozygous for a pLoF variant in the apolipoprotein C3 (APOC3) gene, associated with lipid metabolism. By re-contacting one homozygous proband, researchers went onto identify and recruit six pLoF carriers and seven non-carriers from the same family for detailed physiologic examination. Participants underwent an oral fat load followed by serial blood testing for six hours which showed pLoF homozygotes had lower post-prandial triglyceride excursions. Features from this work that are more broadly applicable within the RbGsv framework include the exploitation of founder populations due to the potential enrichment for highly penetrant large effect variants and the potential to expand recruitment to family members of those identified for recall²⁵⁻²⁸.”

3) I think horizontal pleiotropy will be a much bigger problem than the authors suggest for RbG studies. There is emerging evidence that suggests that pleiotropy is pervasive amongst both GWAS loci and sub-genome-wide significant genetic variants. The authors state that in the paper “horizontal pleiotropy is a complication likely to be more relevant for RbGmv”. This might be true if one has multiple instruments for RbGsv and there is a single variant amongst the multiple variants that has little horizontal pleiotropy. However, if pleiotropy is pervasive (as emerging evidence in the medical genetics community suggests), then its less likely to be able to find the rare instance where horizontal pleiotropy does not occur. Also, I think GRS can also be highly pleiotropic depending on which genetic variants are included. I think a more measured discussion of horizontal pleiotropy, and its impact on RbG would be

helpful.

This is an extremely pertinent point. In immediate address, there are two important arguments to have in mind with a RbG experiment. Firstly, in a RbGsv scenario, we would not advocate any claim of avoiding conventional (or horizontal) pleiotropy – rather, we would suggest that this is an opportunity to really explore the implications of a specific genetic perturbation which has been highlighted through another source of evidence. Secondly, in a RbGmv setting, again we would not suggest that avoiding pleiotropy is a possibility – rather, we would suggest that the very presence of multiple, independent, instruments for a risk factor of interest provides an opportunity to assess and account for the presence or absence of unbalanced horizontal pleiotropy. In this situation, it is arguable that the more independent instruments (that are validated and replicable) for a risk factor of interest, the better the opportunity to rule out a situation of unbalanced horizontal pleiotropy and directional bias in estimates comparing groups defined by genetic difference alone. As said, we agree that these are important issues and as such have extended the discussion on this in the “Future directions ..” section (L407-423).

“In line with more conventional MR analyses based on non-selected population samples, the quality and nature of the genetic variants used for stratum formation will directly affect the ability to draw inference. Population stratification, genuine (or horizontal) pleiotropy and consequent unanticipated instrument properties have been observed elsewhere^{45, 46} and may affect RbG through the invalidation or complication of genetic instruments. Horizontal pleiotropy^{3, 47} specifically is a complication that should be viewed in the context of the type of RbG being undertaken however and is an issue that is pertinent to MR more generally⁴. In the case of RbGsv, whilst pleiotropy may complicate the inference drawn from differences between recalled groups, one of the merits of using RbG for single variants will be to explore the potentially diverse and complicating nature of genetic associations validated for health outcomes. For RbGmv, the situation is different and whilst it is theoretically possible for unbalanced pleiotropy to potentially bias estimates drawn from groups defined by many genetic variants, with increasing numbers of these genetic proxies for complex exposures or risk factors of interest, the likelihood of this problem decreases⁴⁷. This does not remove pleiotropy as a potential complication (indeed it may be unlikely to ever have a single genetic predictor not involved in complex regulation), but it presents RbG as an approach to explore and account for pleiotropy in the right conditions.”

4) The RbG approach would appear to benefit from the class of genetic variants that are most damaging re. protein-coding variation particularly LOF carriers, and then deleterious missense variants. For the RbGmv, I think class of genetic variation is less so since it involves multiple genetic variants with small effects collectively. A discussion of how one would prioritize recalling individuals based on their genotypes (do you look for LOF carriers first?) is helpful.

Thanks for this point – it is of course right to think about the utility of specific variants (or combinations thereof), especially in a RbGsv framework. As mentioned above, we have added a paragraph into the “Exemplars ...” section (L163-179) which charts the potential utility of functional variants in these situations, but there are of course other ways to rank variants for RbG purposes. To this end, resources such as ExAC and others present the possibility to essentially map the human genome (or population based variability therein) for the ranked utility of variation in RbGsv experiments. Coordinated with signatures of trait association, this would allow the obvious extension of the RbG design app through a browser able to describe the likely utility of any variant or group of variants in a RbG context. We have alluded to this in the futures section (L381-392) at the end of the

manuscript, though the execution of this as an extension to the current resource is currently beyond this paper.

“Along with this, there are a series of obvious developments that may enhance the utility and development of RbG as an approach. Resources are already available that present the possibility of mapping the human genome for the ranked utility of variation in RbG experiments. Most pertinent to RbGsv designs, such mapping would allow browsing of genetic regions of interest for evidence of actual or predicted functional variation using best available data (for example, the ExAC database⁴³). Marriage of this to outcome association results and RbG study design parameters would allow pre-emptive assessment of the likely value and performance of a RbG study. In addition to this, other obvious developments include the formalisation of data-driven recall protocols (where the reduction of extremely complex data for non-hypothesis driven association signal discovery is followed by deep exploration of results by genotype) and the testing of population-level opt-out strategies to ensure ethical balance for RbG studies (i.e., that avoid disclosure of genotype status – or likely status – with invitation alone).”

The one element of this type of extension that we remain conscious of is the implications of this type of RbG for the participants involved. Whilst it is relatively straight forward to exercise in vitro analyses to explore a functional variant from a participant point of view (especially in light of new gene editing approaches) and perhaps possible to examine rare (or even personal) variation functionally in patients with a vested interest, it is more difficult to undertake this work ethically in large collections of volunteers. Penetrant and large effect loci (if they are present) need to be handled with care in this context and where they have not already been flagged by family history, accompanied by appropriate support. We also allude to this in the ethics section (L299-311).

“RbG is a potentially powerful research design, but it creates ethical challenges. The RbG approach is inextricably linked to the issue of disclosing potentially sensitive individual results^{36,37} and places an emphasis on transparency and communication with participants. This is of course relates to the nature of both the RbG design and the genetic variation being used to construct the RbG stratum of interest. This is particularly pertinent where potentially penetrant and functional variants are employed in RbGsv designs, but has implications for all forms of RbG. Despite this, there is little published academic work regarding the specific ethical issues in RbG studies. A small body of literature suggests a need for “bottom-up research” to be monitored by an independent governance body³⁸ and that the issues presented with RbG studies are not new but common to those faced by other approaches, such as the use of medical records³⁹. Qualitative data that does exist around this topic compared the experiences of patients (those with the disease of interest) to those of “healthy volunteers” (recalled from a biobank) following their recruitment on the basis of genotype⁴⁰. This research found that, whilst patients expressed “no concerns” about the eligibility criteria, “healthy volunteers” did not always comprehend the study design or why they had been chosen. This led in some cases to participants assuming a degree of meaningfulness to the genetic data that was unwarranted but nevertheless caused them to feel anxious.”

5) Following point 5 above, collection of LOF carriers and deep phenotyping has been discussed in the form of a “human KO project”.
(<http://www.nature.com/nrd/journal/v16/n8/full/nrd.2017.139.html>). Recognition of this specialized form of RbG is warranted.

We assume the reviewer is referring to point (4) above. As mentioned above, we have added a paragraph into the “Exemplars ...” section (L163-179).

6) Following point 6 above, if one focuses on LOF recall, then there may only be N=1 individual to recall. In a N=1 study, the LOF carrier genotype is extremely rare but valuable.

We assume the reviewer is referring to point (5) above. See response to points 4/5 above.

7) I like the measured discussion of RbG not being applicable for all genes and that it depends on the hypothesis. One major area of interest is targeting drug discovery genes which I would welcome a discussion.

We agree that this is a potentially useful area. Again, it is not inconceivable (with initiatives such as the “druggable genome” in operation) to consider listing and potentially ranking RbG genotype selection by relevance to drug discovery. In light of this, we have acknowledged the potential importance of human genetic variation in this by adding to the “Rationale ...” section (L112-122).

“These key features ensure results from RbG studies can be useful in a variety of settings, including in the realm of drug development. For example, data from both GlaxoSmithKline¹⁰ and AstraZeneca¹¹ show that genetic target linkage to disease increases the rate at which drugs are approved. Currently, one of the main sources of genetic support are results from GWAS (for example, those in GWASdb¹² and these seem to be particularly useful in earlier stages of the drug development process¹⁰. However, the influence of genetic support appears to be less strong in progression from Phase III trials to approval¹⁰, suggesting that there is still progress to be made in refining molecular targets. Furthermore, RbG studies may be able to realise the concept of dose-response curves derived from ‘experiments of nature’ described by Plenge et al. (2013)¹³, where naturally occurring mutations can be used to estimate the probable efficacy and toxicity of a drug.”

8) It’s unclear if in the RbGmv design that the authors are proposing utilizing all SNVs in the human genome or just genome-wide significant SNVs for a particular trait or locus for the GRS calculation. If its the former, then this technically is not Mendelian randomization or should be used as causal inference estimation (line 160 - 170) since utilizing all SNVs in the human genome (including ones with tiny effects with P-values~0.05) also includes some SNVs that are definitively associated with the trait of interest (there will be a lot of SNVs with false positives).

We understand and agree with the issue being raised here. Currently, we have developed the idea of RbGmv around the extension of Mendelian randomisation (MR) into a study design space and not as an exercise in genetic prediction. The issues raised above, including unbalanced pleiotropy for replicable genetic associations, are complex already and we would not routinely consider the comprehensive use of all common genetic variation in genomewide predictors as viable applications of MR – even in a RbGmv context. We did not state this explicitly in the previous version of the manuscript as the focus is clearly on the design and potential of RbG, but now allude to the need for care in genetic instrument choice and suggest that the value of a RbGmv is only likely to be as good as the variants used to form the stratum of interest. (L190-210)

“Consistent with conventional MR analyses, the choice of genetic variants for RbGmv studies relies on the ability of genotypic variation to act as a reliable proxy measure for the exposure of interest. Distinct from genetic prediction, this use of multiple genetic variants as markers for modifiable risk (as in more conventional MR designs) requires strong evidence of reliable association. Single genetic variants associated with complex traits or modifiable risk factors often explain only a small proportion of variance in that trait and a strategy employed to try and recover some of the consequent lack of power of single variant analyses is to generate aggregate genetic risk scores (GRSs)²⁹⁻³¹. The use of multiple genetic variants in this way can increase the precision of the causal estimate compared with those derived using separate genetic variants³². In contrast to conventional MR, once a GRS is constructed within the study sample targeted for RbG (usually as the sum of allele dosages at risk variants weighted by their beta coefficients obtained from an independent GWAS for the

exposure of interest), individuals are ranked based on this score, which is then used to stratify participants for recall (Figure 2). Actual selection of individuals from extremes of the GRS will be dependent on the number and frequency of the variants forming the score, their effect and the number of participants (or samples) available. In addition, it should be considered that whilst the average genetic composition of a GRS used to recruit participants will be the same, unlike RbGsv, the precise allocation of genotype will vary participant to participant. Even so, the differences across the genetic stratum will carry the same inferential properties as RbGsv and allow for causal inference concerning the risk factor being instrumented⁶. An example of an RbGsv study is included in **BOX 1**.”

Reviewer #2 (Remarks to the Author):

This well-written paper provides a discussion of technical and ethical aspects of study designs involving recall by genotype, for the purpose of detailed phenotypic or causal inference.

Thank you for your kind appraisal of our manuscript.

The paper would benefit from further clarification on two points:

1-Recall process. The authors appropriately note that recalling by genotype raises ethical concerns about participant harm, based on potential misunderstanding of the genotype for which the participant was recruited. This issue, as they note, is particularly delicate if the study is intended to address substantial uncertainties about the meaning of the genotype. Potential participants need transparent information about study procedures and goals but also communication that minimizes the risk that they will misunderstand and potentially over-estimate the implications of their genotype. They might wish to comment on ways in which empiric studies of communication might help to clarify the most appropriate approaches to address this challenge. As a related issue, and potentially more concerning for some participants, they might comment on the ethical concerns related to obtaining the genotypic information that identifies eligible participants. They note that "large genotyped bioresources" are needed to enable the recall by genotype studies, and note also that increasing use of genomic assessment in clinical studies and increased use of bio repositories will facilitate this approach. They do not discuss the fact that participants may not always be aware of the scope of genomic analysis available from their samples, particularly if genotyping has been done on stored samples or sample obtained as "clinical waste." Ethical issues that call for more attention include the nature of the initial consent process - e.g., how fully have potential participants been informed about the likelihood that they will be approached for studies of this kind? - and the need for researchers to establish what prior information has been provided to participants about the nature of research with their samples, prior to contact. In some instances these issues will have been addressed proactively, but how should researchers address them if they have not?

Since we submitted this article, we have made available in preprint form results of a qualitative research study conducted in the Avon Longitudinal Study of Parents and Children cohort which sought to examine the position of participants with respect to RbG studies (as mentioned above)
(<https://www.biorxiv.org/content/biorxiv/early/2017/04/05/124636.full.pdf>). Following

53 semi-structured interviews conducted with young adult participants of this cohort of ostensibly “healthy volunteers”, researchers reported that few expressed any immediate concerns about being recruited by genotype. The main reasons given for this were the trust that participants had developed over their long-term relationship (more than 20 years) with the study, plus a naturally limited knowledge of genetics and modest interest in reported research outcomes. Whilst this adds to previous research which identified the relationship between researchers and participants as a factor that may influence how much information is provided, with regular study participants perhaps expecting more under the ethical principles of respect and reciprocity (Ravitsky 2006, <http://dx.doi.org/10.1080/15265160600934772>) . As it is yet to be published in a peer-reviewed form we have not included these results in the current draft.

However, in the absence of additional substantive literature on the communication of genetic effects or risks to participants in studies like this, we have added our own experiences and strategic developments to the futures section of the paper. Specifically, we note the previously un-explored problem not of accidental disclosure of genotypes explicitly, but the more subtle effect of increasing the probability of risk variant carriage by simply being invited to a RbG study. This phenomenon has brought about specific changes to the way one approaches RbG studies when global opt-out strategies need to be considered before the invitations to RbG studies are even sent out. This again relies on the nature of the experiment being conducted, but presents an important challenge to the geneticist in terms of communicating genetic effects/risks to the participant. (L387-392)

“In addition to this, other obvious developments include the formalisation of data-driven recall protocols (where the reduction of extremely complex data for non-hypothesis driven association signal discovery is followed by deep exploration of results by genotype) and the testing of population-level opt-out strategies (i.e., that avoid disclosure of genotype status – or likely status – with invitation alone) to ensure ethical balance for RbG studies.”

We have also added a discussion around the likely different types of resources that could be used and how this might impact on issues such as consent to the “Ethical considerations ...” section, highlighting the difference between purpose-built RbG bioresources and other prospective cohort studies. (L323-342)

“Related to the nature of the cohort is the extremely important issue of consent and the provision for re-contact of participants within the informed consent process of the original study^{40,42}. Whilst there are a number of “purpose-built” RbG resources such as The Oxford Biobank, the Exeter 10,000 (EXTEND), the East London Genes & Health (ELGH) and the Extended Cohort for E-health, Environment and DNA (EXCEED) projects whose consent processes deal explicitly with the issue of RbG, in many cases researchers will be looking to recruit from cohort studies established for more general epidemiological research. Therefore, in the event that a network approach to RbG studies is initiated (as described below), careful consideration will need to be given to the extent to which consent and disclosure policies can and should be aligned across studies versus the tailoring of approaches to account for the varied nature of the cohorts involved.”

2-Appropriateness of controls/biased sampling- The authors discuss selection of comparison groups by genotype, a methodological issue which is fundamental for this kind of research. They also comment in passing about pari-matching for age, sex and BMI. However, given the many indications we have of gene-environment interactions and of the impact of social environments on health outcomes that are also influenced by genotype, it would seem that researchers should also consider other variables relevant to their study question. This

methodological point does not seem to have been considered. it may be particularly important given a likely socioeconomic bias in most large genotyped resources.

We thank the reviewer for this valuable point. We discussed this very point at length in the formulation of the manuscript for two reasons. Firstly, with sufficient sample size and in the absence of bias by genotype (generated potentially by population stratification), the value in recruiting by genotype comes from the essentially random distribution of genotype to other genotypes and other confounding factors – this is the core property of genetic variation exploited by conventional MR. In this context, there should be theoretically no need for any matching or balancing after recruitment by genotype alone and in the theoretical absence of the inferential complications of pleiotropy, differences between the recalled groups can be attributed to genotype alone. Secondly, and counter to this, matching according to some specified variables may serve to increase the efficiency of a RbG study in certain conditions. To this end, gene*environment interactions may fit in here, however, few have been reliably shown to date. Currently we outline the potential utility of matching in a RbG setting, though wish to maintain the parallel of RbG to MR and not overcomplicate this further in the manuscript. We have revised the relevant paragraph and acknowledged the potential role of G*G and G*E interactions. (L283-287).

“Other situations that may warrant consideration include gene*environment and gene*gene interaction, though the evidence for consistent examples of these in the literature has been limited to date. Importantly, there remains a danger that such manoeuvres can exacerbate the potential for particular types of study bias³⁵ and the pros and cons of these decisions need to be weighed carefully in study design.”

Reviewer #3 (Remarks to the Author):

In this paper, Corbin and colleagues discuss the motivation for and characteristics of a Recall-by-Genotype (RbG) study design for following up genetic association results. They describe two settings for employing a RbG – one to follow-up single variant associations and a second that utilizes multiple variants in a risk score type setting. As acknowledged by the authors, the underlying concept behind the RbG design is not new, having been used in epidemiology for many years – that is, recruitment by exposure status. The concept is not particular new for genetic studies either. The authors’ intent in this paper is to provide a conceptual framework to this approach.

Overall, I think the motivations for the RbG studies are relatively straightforward and are appropriately spelled out in this article.

Thank you for your kind appraisal of our manuscript.

There are a few areas where I think the authors can elaborate:

1. Founder populations constitute a particularly strong niche for RbG studies because of the potential enrichment for highly penetrant, large effect variants that can provide insights into biology. For example, in the Amish population the RbG design has been used for some time to gain biological insights by calling back selected subjects for more detailed phenotyping (e.g., APOC3: Pollin et al., Science, 2008; ABCG8: Horenstein et al Arteriosler Thromb Vasc Biol, 2013; LIPE: Albert et al., New Engl J Med, 2014; COL1A2: Daley et al., J

Bone Miner Res, 2009; SLC30A8: Maruthur et al., Diabetologia, 2015) An added feature of the RbG design that the authors might note is the potential not just to call back prior study participants with genotypes of interest, but also to expand recruitment around family members of these individuals.

This is a very good point and one we are keen to mention directly in the paper. We have added a paragraph about pLoF variants in “Exemplars ...” section, which also addresses the point re. usefulness of founder populations for RbG more generally, using the examples provided (L176-179).

“Features from this work that are more broadly applicable within the RbGsv framework include the exploitation of founder populations due to the potential enrichment for highly penetrant large effect variants and the potential to expand recruitment to family members of those identified for recall²⁵⁻²⁸.”

2. Consent issues and ethical issues: One point that is not explicitly made is that there may be IRB barriers for some studies from recalling individuals based on genotype since this requires disclosure of genotype information, and study subjects may not have been

In the previous draft, we have attempted to flag the potential problems faced with RbG concerning disclosure. As mentioned above, we also explicitly consider not only direct disclosure, but also probabilistic inference through invitation alone. In addition to this, we have expanded the “Ethical considerations ...” section including a discussion of the likely different consents depending on study (resource) design. (L332-342)

Ethics section:

“The very nature of RbG designs highlight a central tension between avoiding the possibility of participant harm through revealing unwanted or misunderstood information and being open and clear when explaining how and why participants are being recruited into studies^{36,37}. In healthy volunteers, it is unlikely that the genetic information used for recruitment to most RbG studies will be either immediately clinically valid or useful, as the precise function of the genetic characteristics will presumably be unknown. However, this does not diminish the need to clearly communicate the study protocol to participants and why they, specifically, have been recruited. To this end, the issue of direct or unwanted indirect disclosure of genotype is of great importance in this type of study. It is of course possible to envisage a situation whereby a threshold of clinical relevance obtained through an RbG study is not reached, but the genetic information could still be of interest to the participant. The employment of sensible mechanisms for assessment of data quality and routes for appropriate feedback (as considered in detail for sequencing studies elsewhere)⁴¹ will clearly be the accepted mode for RbG studies with large effects. However, the issue of addressing a specific genotype-driven effect does serve to illustrate a key advantage of RbG studies over less hypothesis-driven genomic research. It is potentially easier to anticipate the nature of findings for a given recall stratum and therefore the potential relevance of those findings to participants^{36,37}.”

Related to the nature of the cohort is the extremely important issue of consent and the provision for re-contact of participants within the informed consent process of the original study^{40,42}. Whilst there are a number of “purpose-built” RbG resources such as The Oxford Biobank, the Exeter 10,000 (EXTEND), the East London Genes & Health (ELGH) and the Extended Cohort for E-health, Environment and DNA (EXCEED) projects whose consent processes deal explicitly with the issue of RbG, in many cases researchers will be looking to recruit from cohort studies established for more general epidemiological research. Therefore, in the event that a network approach to RbG studies is initiated (as described below), careful consideration will need to be given to the extent to which consent and disclosure policies can and should be aligned across studies versus the tailoring of approaches to account for the varied nature of the cohorts involved.”

Reviewer #1 (Remarks to the Author):

The new additions and revisions have improved the manuscript particularly on expanding discussion around pleiotropy on a RbG experiment and utility of specific genetic variants as appropriate genetic instruments. My previous comments have been appropriately addressed.

If the authors would like to remain current with the most recent literature, there has been an interesting preprint published after submission that suggests that the reason focusing on the very top percentile of individuals with genetic risk scores would be useful for follow-up is that their genetic risk to complex disease is almost as strong as a single Mendelian mutation (<https://doi.org/10.1101/218388>). The authors could consider briefly mentioning this in the GRS section, although it is understood that this preprint was published just very recently.

Reviewer #2 (Remarks to the Author):

This paper provides an interesting discussion of an important methodology but the responses related to consent and gene-environment interaction are quite limited. On consent, perhaps noting the revised discussion is the best that can be offered - i.e., noting the value of prospective coverage of recall by genotype when it is anticipated, and dealing carefully with the re-contact process when it is not. But this does seem to be an area where more research is needed - in particular, given the data cited by the authors, additional research should perhaps be directed toward understanding the participant experience in situations where there is not a long-term association between participant and the research team.

On gene-environment interaction, the authors suggest that biases are averted by random distribution of genotype. However, for social mediators of health risk, random distribution of genotype is unlikely, given the powerful associations between social position and geographic ancestry in many populations.

Reviewer #3 (Remarks to the Author):

The authors have adequately addressed the 2 specific issues I raised in my initial review.

REVIEWERS' COMMENTS:

Reviewer #1 (Remarks to the Author):

The new additions and revisions have improved the manuscript particularly on expanding discussion around pleiotropy on a RbG experiment and utility of specific genetic variants as appropriate genetic instruments. My previous comments have been appropriately addressed.

If the authors would like to remain current with the most recent literature, there has been an interesting preprint published after submission that suggests that the reason focusing on the very top percentile of individuals with genetic risk scores would be useful for follow-up is that their genetic risk to complex disease is almost as strong as a single Mendelian mutation (<https://doi.org/10.1101/218388>). The authors could consider briefly mentioning this in the GRS section, although it is understood that this preprint was published just very recently.

Thank you for your further comments, we are pleased that you are satisfied with our responses. With regards to the preprint article that you mention, this is indeed of relevance to our work and we thank you for bringing it to our attention. However, given the journal's policy on referencing preprints we have decided not to reference this article in the manuscript.

Reviewer #2 (Remarks to the Author):

This paper provides an interesting discussion of an important methodology but the responses related to consent and gene-environment interaction are quite limited. On consent, perhaps noting the revised discussion is the best that can be offered - i.e., noting the value of prospective coverage of recall by genotype when it is anticipated, and dealing carefully with the re-contact process when it is not. But this does seem to be an area where more research is needed - in particular, given the data cited by the authors, additional research should perhaps be directed toward understanding the participant experience in situations where there is not a long-term association between participant and the research team.

On gene-environment interaction, the authors suggest that biases are averted by random distribution of genotype. However, for social mediators of health risk, random distribution of genotype is unlikely, given the powerful associations between social position and geographic ancestry in many populations.

Thank you for your further comments.

On gene-environment interaction, we have altered the text to be more explicit about the potential issues faced in this situation (L477-489):

*“Other situations that may prompt refinement of the basic RbG design include instances of gene*environment and gene*gene interaction. Though the evidence for consistent examples of these in the literature has been limited to date, in the presence of a gene*environment interaction, for example, the assumption that genotype is orthogonal to all potential confounders may be invalidated due to associations between socioeconomic status and geographic ancestry. Importantly, there remains a danger that efforts to balance*

or match samples can exacerbate the potential for particular types of study bias³⁵ and the pros and cons of these decisions need to be weighed carefully in study design.”

On the ethics of consent, since the editor has permitted us to discuss a preprint, we have extended the discussion in the first two paragraphs of the ‘Ethical considerations of RbG’ section (L529 – 544 in tracked version) to include the remarks we made in response to your last review. This includes bringing across some of the comments made in the rebuttal and referencing both the preprint we mention and another published article (Ravitsky 2006). We hope these edits meet with your approval.

Reviewer #3 (Remarks to the Author):

The authors have adequately addressed the 2 specific issues I raised in my initial review.

Thank you, we are pleased that you are satisfied with our responses.